# Is the Pendulum of Antimicrobial Drug Resistance Swinging Back after COVID-19?

**DOI:** 10.3390/microorganisms10050957

**Published:** 2022-05-02

**Authors:** Francesca Serapide, Angela Quirino, Vincenzo Scaglione, Helen Linda Morrone, Federico Longhini, Andrea Bruni, Eugenio Garofalo, Giovanni Matera, Nadia Marascio, Giuseppe Guido Maria Scarlata, Claudia Cicino, Alessandro Russo, Enrico Maria Trecarichi, Carlo Torti

**Affiliations:** 1Unit of Infectious and Tropical Diseases, Department of Medical and Surgical Sciences, “Magna Graecia” University, 88100 Catanzaro, Italy; francescaserapide@gmail.com (F.S.); helen.morrone@gmail.com (H.L.M.); a.russo@unicz.it (A.R.); em.trecarichi@unicz.it (E.M.T.); torti@unicz.it (C.T.); 2Unit of Clinical Microbiology, Department of Health Sciences, “Magna Graecia” University, 88110 Catanzaro, Italy; quirino@unicz.it (A.Q.); gm4106@gmail.com (G.M.); nadiamarascio@gmail.com (N.M.); giuseppescarlata93@gmail.com (G.G.M.S.); claudiacicino@gmail.com (C.C.); 3Unit of Intensive Care, Department of Medical and Surgical Sciences, “Magna Graecia” University, 88100 Catanzaro, Italy; flonghini@unicz.it (F.L.); andreabruni@unicz.it (A.B.); eugenio.garofalo@unicz.it (E.G.)

**Keywords:** COVID-19, antimicrobial resistance, MDR

## Abstract

The COVID-19 pandemic may have had an effect on antimicrobial resistance. We compared the prevalence of ESKAPE multidrug-resistant (MDR) bacterial infections in COVID-19 affected/unaffected patients admitted to intensive care units (ICU) or infectious disease units at the “Mater Domini” University Hospital of Catanzaro between 1 March 2020 and 31 July 2021. Moreover, an analysis of MDR rates in ICU comparing the pre-pandemic period with the pandemic period was performed, and the possible consequence on in-hospital mortality was explored. One hundred and eighty-four ESKAPE isolates were analyzed from 362 SARS-CoV-2 positive and 199 negative patients. In total, 116 out of 171 Gram-negative isolates were classified as MDR, and a higher frequency was observed in COVID-19 compared with non-COVID-19 patients (74.2% vs. 60.3%; *p* = 0.052). A higher rate of MDR ESKAPE bacteria was observed in COVID-19 patients admitted to the ICU compared with COVID-19 unaffected patients admitted to the same ward in 2019 (88% vs. 80.4%; *p* = 0.186). *Acinetobacter baumannii* was the main pathogen in COVID-19 patients (58.7%), where it was the most frequent cause of bloodstream infection with the highest mortality rate (68.7%). Increase in MDR appeared to be associated with COVID-19 but only in the ICU setting. *Acinetobacter baumannii* was associated with the risk of death, indicating the importance of implementing infection control measures urgently.

## 1. Introduction

Our group recently published results showing a favorable trend in the reduction of antibiotic resistance from 2015 to 2019 following a persuasive antimicrobial stewardship program [1]. From 2020, the national healthcare system endured considerable pressure due to the SARS-CoV-2 pandemic, with several reports showing data that evidenced an increased risk of spread of antibiotic resistance though it was not invariably confirmed [2,3]. This negative trend could be explained by many factors that take place in COVID-19 settings. For example, SARS-COV-2-infected patients are admitted to dedicated areas in which isolation precautions are difficult to apply due to structural and human resource limitations [4]. In addition, patients with severe COVID-19 pneumonia often require mechanical ventilation and multiple catheter insertions, including those for extracorporeal membrane oxygenation (ECMO) [5]. The length of hospital stay in severe COVID-19 tends to be longer than for other acute infectious diseases such as influenza, exposing the patients to a greater risk of nosocomial infections [6]. Moreover, despite poor evidence regarding the matter, a significant number of patients is treated with empiric broad-spectrum antibiotic therapy, thus increasing the risk of developing infections caused by multidrug-resistant (MDR) pathogens [7]. Lastly, the use of drugs targeting cytokines such as IL-6 in patients with the so-called COVID-19-related “cytokine storm” also may constitute a risk factor for superinfection by compromising the immune system [8].

We, therefore, extended our above-mentioned analysis to the pandemic era (i.e., from 1 March 2020 to 31 July 2021), adopting the same methods [1] to explore whether an increase in ESKAPE multi-drug resistant (MDR) bacterial infections has occurred and to define the possible consequences of this phenomenon on in-hospital mortality. COVID-19 patients admitted to an intensive care unit (ICU) or infectious disease unit were compared with non-COVID-19 patients admitted to the same units. A focus on in-hospital mortality analysis was performed only in patients with monomicrobial bloodstream infections (BSI) to provide a univocal interpretation.

## 2. Materials and Methods

We analyzed all initial ESKAPE bacterial strains isolated from blood, urine, and respiratory samples in COVID-19 affected/unaffected patients admitted to the “*Mater Domini*” University Hospital of Catanzaro between 1 March 2020 and 31 July 2021. The diagnosis of SARS-CoV-2 infection was confirmed by RT-PCR. Bacterial coinfections or colonizations were detected by standard culture isolation from respiratory, blood, or urine samples. Surveillance nasal, oropharyngeal, and rectal swabs were excluded. Pure bacterial cultures and antibiotic susceptibility testing were performed using the automated VITEK^®^ system (BioMérieux, Marcy-l'Étoile, France), including appropriate quality controls. Susceptibility to antibiotics was interpreted based on the European Committee on Antimicrobial Susceptibility Testing (EUCAST) breakpoints [9]. Microbial isolates displaying intermediate susceptibility to antibiotics were merged with those showing resistance in order to provide a worst-case scenario to increase detection of possible circulation of bacteria not optimally responding to treatment, as already performed in our previous study [1]. We compared the incidence of MDR bacteria isolated from the above-mentioned clinical samples in COVID-19 positive and negative patients. A comparison between the pre-pandemic period and pandemic period was made in the ICU ward, and in-hospital mortality was correlated with MDR and bacteria isolated a single species from hemocultures.

The study was conducted using retrospectively collected and anonymized data according to the Declaration of Helsinki and principles of good clinical practice. According to the Italian legislation (GU Serie Generale no. 76 31/3/2008), due to the retrospective nature of the study and considering the absence of any demographic and clinical data of the patients, only notification was due to the Ethical Committee, which was sent on 22 March 2019.

## 3. Results

Overall, 184 ESKAPE isolates were analyzed from 362 SARS-CoV-2 positive and 199 SARS-CoV-2 negative patients. Most pathogens were isolated from the bloodstream (60/184 = 32.6%) or respiratory tract (94/184 = 51.1%). Out of these, 35/60 (58.3%) pathogens from the bloodstream and 45/94 (47.9%) from respiratory samples were isolated from COVID-19 patients. *Acinetobacter baumannii* and *Klebsiella pneumoniae* were the most frequent species in COVID-19 compared with non-COVID-19 patients: 46/99 (46.5%) vs. 32/85 (37.6%) and 33/99 (33.3%) vs. 26/85 (30.6%), respectively.

Table 1 shows the number of MDR and non-MDR bacterial isolates, stratified by samples and hospital unit, in COVID-19 and non-COVID-19 affected patients.

### 3.1. MDR and Non-MDR Bacterial Isolates

One hundred and sixteen out of 171 (67.8%) Gram-negative isolates were classified as MDR pathogens, and a higher frequency was observed in COVID-19 compared with non-COVID-19 patients (69/93 = 74.2% vs. 47/78 = 60.3%; *p* = 0.052).

A higher rate of MDR ESKAPE bacteria was observed in COVID-19 patients admitted to the ICU compared with COVID-19 unaffected patients admitted to the same ward in 2019 (66/75 = 88% vs. 74/92 = 80.4%); however, statistical significance was not achieved (*p* = 0.186).

The lowest rate of MDR ESKAPE bacteria was found in patients admitted to the non-COVID-19 ICU in the COVID-19 pandemic period (48/79 = 60.8%), compared to both non-COVID-19 patients in 2019 (*p* = 0.0045) and to COVID-19 patients in the pandemic period (*p* = 0.0001).

Figure 1 shows the percentage of multidrug-resistant bacteria of the ESKAPE group in the intensive care unit before and during the COVID-19 pandemic period.

### 3.2. Distribution of Bacterial Isolates in COVID-19 and Non-COVID-19 Affected Patients

*Acinetobacter baumannii* was the main pathogen accounting for MDR ESKAPE etiologies, increasing from 33.9% among ESKAPE bacteria in non-COVID-19 patients in 2019 to 58.7% in COVID-19 patients in the period March 2020 to July 2021 (*p* = 0.0001; Figure 1).

### 3.3. In-Hospital Mortality: Focus on Bloodstream Infection

Regarding in-hospital mortality, 50 monomicrobial blood cultures were analyzed, 29 (58%) of them taken from COVID-19 patients. A greater frequency of Gram-negative MDR bacteria was reported in COVID-19 patients (23/25 = 92%) compared with unaffected ones (12/18 = 66.6%; *p* = 0.03). Rates of in-hospital mortality after BSI were found, in decreasing order, in the following groups: COVID-19 patients in ICU when etiology was *Acinetobacter baumannii* (11/16 = 68.7%), COVID-19 patients in ICU when other etiologies were detected (5/8 = 62.5%), non-COVID-19 patients in ICU with *Acinetobacter baumannii* causing BSI (5/9 = 55.5%), and, lastly, non-COVID-19 patients in ICU with other etiologies for BSI (3/11 = 27.3%).

## 4. Discussion

This study emphasizes the negative impact of the COVID-19 pandemic on antimicrobial resistance trends in ICU settings in which a high rate of ESKAPE Gram-negative MDR bacterial infections was detected (88%). It seems that the pendulum is swinging back in the ICU, where COVID-19 may have worsened the antimicrobial resistance issue, while in the non-COVID-19 wards, we observed a continuing trend toward a reduction in MDR prevalence in line with what has been found in the pre-pandemic period [1]. This result may be explained by a “forgiveness effect” of the persuasive stewardship measures in non-COVID-19 areas previously implemented [1], which may have been favored by a reduction in the number of non-COVID-related hospitalizations; thus, the lower rate of MDR infection may also reflect the reduction in bed occupation in non-COVID-19 wards [10,11,12].

The causative agent *Acinetobacter baumannii* appeared to pose the greatest threat to COVID-19 patients admitted to ICUs. In fact, all *Acinetobacter baumannii* isolates were MDR, and their prevalence was higher in COVID-19 affected patients compared with unaffected ones. The highest prevalence was found in blood culture, which is particularly alarming considering the negative outcome of patients with sepsis from *Acinetobacter baumannii* [13,14]. Indeed, the raw survival analysis in our work appears to confirm that both COVID-19 pathological condition and positive blood culture for *Acinetobacter baumannii* are correlated with a greater risk of death in patients analyzed in the pandemic era [15,16].

It is difficult to understand the causes of the increase in bacterial species that are multiresistant to antibiotics in patients with COVID-19 admitted to ICU, but we would like to raise attention to the following points: (i) both the intrahospital transmission of these bacteria and a nonoptimal antibiotic stewardship may have exerted an important role in raising antimicrobial resistance rates [17,18]; (ii) COVID-19-dedicated ICU appeared an important hotspot for the spread of MDR pathogens, probably for some specific peculiarities that should be addressed: patient overcrowding, insufficient personnel, suboptimal adherence to standard precautions due to the use of individual protective equipment, insufficient rapid isolation of patients with MDR colonization at admission [4], and these factors may have concurred to determine several outbreaks that occurred as reported also by other authors [19,20]; (iii) a cross-transmission of MDR bacteria may have occurred through importation from other hospitals since “*Mater Domini*” teaching hospital is the reference center of Calabria Region for care of patients with severe respiratory infections; (iv) despite poor evidence [7], antibiotic therapy is often prescribed in COVID-19 patients both in in-hospital and in outpatient settings, thus creating a collateral damage by limiting available antibiotic choices [21,22]. Although restriction in prescribing antibiotic therapies is important to avoid the selection of MDR bacteria, the severe clinical conditions of patients admitted to ICU and immunodeficiency associated both with COVID-19 by itself and with immunomodulating therapies prescribed make it difficult to apply this rule in most patients [2]. It is, therefore, important to implement more specific protocols of antimicrobial stewardship for COVID-19 patients, particularly for those admitted to the ICU.

The limited number of cases included in this study requires an extension of our observations to a larger number of patients. Moreover, neither clonal analysis nor resistant gene molecular testing was performed on the isolated strains. Lastly, in line with our previous analysis [1], intermediate resistance to antibiotics was considered indicative of a lack of susceptibility to them to provide a more sensitive outcome. However, we do not feel that a different categorization would change the results because only 19 isolates overall in the COVID-19 period showed intermediate susceptibility for any drugs tested.

In conclusion, COVID-19 and antimicrobial resistance appeared to be parallel and interlaced health emergencies [23], and an increase in MDR was directly associated with in-hospital mortality (especially for BSI caused by *Acinetobacter baumannii*). Stewardship programs and infection control interventions should, therefore, be urgently implemented and adapted to the specific setting of the ICU wards dedicated to COVID-19 patients.

## Figures and Tables

**Figure 1 microorganisms-10-00957-f001:**
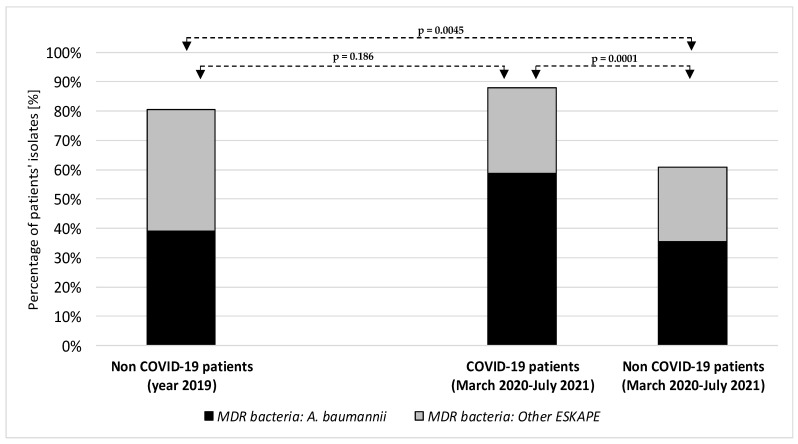
Percentage of multidrug-resistant bacteria of the ESKAPE group in the intensive care unit before and during the COVID-19 pandemic period. *A. baumannii*: *Acinetobacter baumannii; ESKAPE*: *Enterococcus faecium, Staphylococcus aureus, Klebsiella pneumoniae, Pseudomonas aeruginosa, Enterobacter species*.

**Table 1 microorganisms-10-00957-t001:** Distribution of bacterial isolates by hospital units and samples in COVID-19 and non-COVID-19 patients.

Category		SARS-CoV-2 Positive Patients				SARS-CoV-2 Negative Patients					
		Infectious Disease Unit		Intensive Care Unit		Infectious Disease Unit		Intensive Care Unit			
Bacterial Isolates	Samples										
		2020–2021		2020–2021		2020–2021		2019		2020–2021	
		Non-MDR	MDR	Non-MDR	MDR	Non-MDR	MDR	Non-MDR	MDR	Non-MDR	MDR
*Enterococcus faecium*	Urine	0	1	0	1	0	0	0	2	0	3
	Respiratory	0	0	0	0	0	0	0	0	0	0
	Blood	0	1	0	2	0	0	0	2	0	1
*Staphylococcus aureus*	Urine	0	0	0	0	0	0	0	0	0	0
	Respiratory	0	0	0	0	1	0	2	1	2	0
	Blood	1	0	0	0	0	0	4	1	2	0
*Klebsiella pneumoniae*	Urine	5	4	0	0	0	1	1	1	0	3
	Respiratory	1	0	2	11	0	1	1	11	5	8
	Blood	0	2	0	8	0	0	0	9	3	5
*Acinetobacter baumannii*	Urine	0	0	0	1	0	0	0	3	0	3
	Respiratory	1	0	0	24	0	1	0	19	3	14
	Blood	1	0	0	19	0	0	0	14	0	11
*Pseudomonas aeruginosa*	Urine	5	0	0	0	0	0	3	0	1	0
	Respiratory	0	0	4	0	1	0	4	5	11	0
	Blood	0	0	1	0	0	0	0	1	2	0
*Enterobacter* species	Urine	2	0	0	0	0	0	0	0	0	0
	Respiratory	0	0	2	0	0	0	3	3	2	0
	Blood	0	0	0	0	1	0	0	2	0	0

## Data Availability

The data that support the findings of this study are available on request to the corresponding author (V.S.).

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
