# Peer review of "Is the Pendulum of Antimicrobial Drug Resistance Swinging Back after COVID-19?"

_microorganisms, 2022, doi:10.3390/microorganisms10050957_

Round 1

Reviewer 1 Report

First of all, I congratulate the authors on a very interesting and especially beneficial work that shows the relationship of Covid-19 to antimicrobial resistance (AMR). The information presented is undoubtedly very important and highlights the dangers of AMR, which has increased in connection with the Covid-19 pandemic. However, I have the following essential comments on the work and I recommend that the authors consider these and modify the manuscript based on them.

Major comments

  • The authors report that the bacterial pathogens causing the infections were analyzed. Unfortunately, the definition of these infections is very vague. The text states "Bacterial coinfections were detected by standard culture isolation from respiratory, blood and urine samples“. I personally consider this to be insufficient and the main weakness of the manuscript. Positive bacterial cultivation does not necessarily mean an ongoing bacterial infection. Infection is defined primarily by clinical parameters and not by microbiological findings. For example, VAP can be defined by clinical and biochemical parameters independently of the isolation of the bacterial agent, which may or may not be identified. And this is the weak point.
  • How have possible contamination of the tracheal secretion with the upper airway bacterial microflora been ruled out? I fully understand the authors' intention, and if bacterial coinfection has not been clearly demonstrated in patients, I recommend reworking the manuscript on "Comparing the incidence of MDR bacteria isolated from the above mentioned clinical materials in Covid-19 positive and negative patients."
  • The authors state that nasal samples were excluded. And what about oropharyngeal samples?
  • The authors state that AMR was detected by conventional methods. I do not consider this to be sufficient and please provide a precise methodology, including information on quality control.
  • Personally, I do not agree with the authors' interpretation in the case of intermediate susceptibility, this is interpreted as resistance in the manuscript. However, according to EUCAST, this category means susceptible - increased exposure. A microorganism is categorized as "susceptible, increased exposure" when there is a high likelihood of therapeutic success because exposure to the agent is increased by adjusting the dosing regimen or by its concentration at the site of infection. Why did the authors use a different classification?
  • I recommend using the full names of the bacteria, not their abbreviations (see Table 1). Or at least explain the abbreviations. These abbreviations may not be obvious to all readers.
  • In the case of Acinetobacter baumannii, I recommend adding a possible explanation of their high frequency by clonal spreading. Do the authors have the results of determining the clonality of isolated strains? If so, I recommend reporting these results.

Reviewer 2 Report

The manuscript of Serapide et al. analysed the pathogen occurrance in patients in COVID-19 ward and non-COVID-19 wards at a University Hospital in Italy in a short-lenght article. It is well suiting to the special issue.and the topic is important still and is well documented. The title may attract the attention of the readers. However, the authors may discuss their data in the Discussion section in more depth that how the more serious COVID-19 ward data compare to the controls with respect to the title and also make conclusions.

Special comment

Lane 103: statistical significance    
